# Black-Box Tuning of Vision-Language Models with Effective Gradient Approximation

**Zixian Guo**[1,2*]   **Yuxiang Wei**[1]   **Ming Liu**[1]   **Zhilong Ji**[2]   **Jinfeng Bai**[2]   **Yiwen Guo**[4]   **Wangmeng Zuo**[1,3(✉)]

[1]Harbin Institute of Technology   [2]Tomorrow Advancing Life   [3]Pazhou Lab, Guangzhou   [4]Independent Researcher

zixian_guo@foxmail.com   yuxiang.wei.cs@gmail.com   csmliu@outlook.com   zhilongji@hotmail.com

jfbai.bit@gmail.com   guoyiwen89@gmail.com   wmzuo@hit.edu.cn

## Abstract

Parameter-efficient fine-tuning (PEFT) methods have provided an effective way for adapting large vision-language models to specific tasks or scenarios. Typically, they learn a very small scale of parameters for pre-trained models in a white-box formulation, which assumes model architectures to be known and parameters to be accessible. However, large models are often not open-source due to considerations of preventing abuse or commercial factors, hence posing a barrier to the deployment of white-box PEFT methods. To alleviate the dependence on model accessibility, we introduce collaborative black-box tuning (CBBT) for both textual prompt optimization and output feature adaptation for black-box models. Specifically, considering that the backpropagation gradients are blocked, we approximate the gradients of textual prompts by analyzing the predictions with perturbed prompts. Secondly, a lightweight adapter is deployed over the output feature of the inaccessible model, further facilitating the model adaptation process. Empowered with these designs, our CBBT is extensively evaluated on eleven downstream benchmarks and achieves remarkable improvements compared to existing black-box VL adaptation methods. Our code will be made publicly available.

## 1  Introduction

Large-scale vision-language (VL) models (Radford et al., 2021; Jia et al., 2021; Li et al., 2021; Yao et al., 2021; Alayrac et al., 2022; Yuan et al., 2021) have demonstrated remarkable performance in a wide range of applications. Various model fine-tuning methods have been proposed to exploit the potential of pre-trained VL models for downstream vision (Zhou et al., 2022b; Lu et al., 2022b; Wang et al., 2022; Sun et al., 2022c; Zhang et al., 2022; Wortsman et al., 2022; Li et al., 2023) and natural language processing (Lu et al., 2022a; Yan

et al., 2022) tasks. Most existing methods conduct parameter-efficient fine-tuning (PEFT (Houlsby et al., 2019)), which updates a tiny fraction of the model parameters or introduces a small number of extra parameters for tuning, in order to transfer pre-trained knowledge in a computation- and data-efficient manner.

Although impressive improvements have been achieved, standard PEFT methods need to pass signals forward and backward through the entire pre-trained model to update the parameters, which relies on the availability of the architecture, parameters, and even the inference source code of the model. Nevertheless, the trend of building machine learning models as a service leads to many proprietary services that only provide an API interface for model inference, *e.g.*, ChatGPT, Bard, and GPT-4, where the parameters and inference code of the models are not open-source due to commercial or safety considerations. Under such black-box circumstances, existing PEFT methods can hardly be adopted. Thus, it is worthwhile to develop methods that can tune pre-trained VL models in a black-box setting. Moreover, in the era of large foundation models, running super large pre-trained models on local devices can be very costly as the scale of the pre-trained model has constantly increased. Although existing PEFT methods restrict learnable parameters to a fairly small scale, it is still a burden to accommodate models with billions of parameters in limited computing resources for most users.

To tackle these problem of tuning black-box VL models, there exist a few very recent efforts. For instance, BlackVIP (Oh et al., 2023) pioneered black-box prompting for VL models by learning an asymmetric autoencoder-style coordinator with a zeroth-order optimization to modify visual prompts in the pixel space. However, modifying prompts in the large pixel space causes inefficiency and the method requires up to 9k parameters in the coordinator to achieve the goal. Besides, the performance

---

*Work done when Zixian Guo was an intern at TAL.

of their visual prompts is subject to the diverse semantic features of a well-trained generative self-supervised learning model. Even so, the method demonstrates limited performance improvements after prompting, showing that prompt tuning in the black-box setting is very challenging.

In this paper, we propose a collaborative black-box tuning method dubbed CBBT for tuning pre-trained VL models and adapting them to downstream tasks. Unlike in BlackVIP (Oh et al., 2023), we learn the prompt for the textual input instead of images, and we adapt the visual features using an adapter. The basic idea is illustrated in Fig. 1.

A query-efficient approximation method (Wierstra et al., 2014) is used to estimate the gradients and optimize the textual prompt with the black-box pre-trained VL model, from which true gradients are not accessible. Specifically, we query the model with randomly perturbed prompts and then summarize the change in model prediction loss to estimate the gradient of learnable parameters (*i.e.*, the prompts). We equip single-step gradient optimization with information from history updates via a momentum strategy, which leads to faster convergence and better results.

Under the circumstance where the output features are available for the pre-trained VL models, we further adapt the visual features by introducing a lightweight adapter module. As demonstrated in Fig. 1, the visual adapter can be learned effortlessly by supervised learning, without having knowledge of the pre-trained VL backbone.

With the joint optimization of the textual prompt and the visual adapter, our CBBT achieves significant model adaptation performance. To evaluate its effectiveness, we conduct extensive experiments on eleven downstream benchmarks, showing superior performance compared to existing black-box VL adaptation methods.

The main contributions of this work can be summarized as follows:

- We advocate textual prompting for adapting pre-trained black-box VL models to downstream tasks. Satisfactory prompt tuning results are obtained with an effective gradient approximation algorithm.

- We expedite the tuning process by utilizing history updates as beneficial information for each optimization step, which brings about accelerated convergence and better results.

- We adapt the visual features jointly with the textual prompt when output features are available. The comprehensive comparison shows that our method achieves state-of-the-art performance compared to other black-box tuning approaches.

## 2 Related Work

**Black-box Prompt Tuning for Large Language Models.** BBT (Sun et al., 2022b) adopts derivative-free optimization using covariance matrix adaptation evolution strategy (CMA-ES) (Hansen et al., 2003) to optimize the prompt in a low-dimensional intrinsic subspace. With this method, the adaptation of large language models works well on natural language tasks, surpassing even the white-box prompting performance. BBTv2 (Sun et al., 2022a) further enhances the capacity of BBT by using deep prompt tuning. BDPL (Diao et al., 2022) tunes a set of discrete prompts for language models by modeling the choice of words in the prompt as a policy of reinforcement learning, and a variance-reduced policy gradient estimator (Williams, 1992; Dong et al., 2020; Zhou et al., 2021) is used to optimize the discrete prompt based on loss value.

**Black-box Adaptation for VL Models.** To the best of our knowledge, BlackVIP (Oh et al., 2023) is the first work to tackle black-box tuning problem of pre-trained VL models. It designs an asymmetric autoencoder-style coordinator to generate input-dependent image-shaped visual prompts and optimize the coordinator by zeroth-order optimization using simultaneous perturbation stochastic approximation (SPSA) (Spall, 1992, 1998, 1997). However, the improvement brought by this method (after visual prompting) is relatively limited compared to the baseline, *i.e.*, the pre-trained CLIP (Radford et al., 2021). LFA (Ouali et al., 2023) liberalizes the regimes of black-box models by assuming pre-computed features from pre-trained backbones are accessible. They optimize a projection layer for a better alignment between pre-computed image features and class prototypes by a multi-stage procedure. They first solve the orthogonal procrustes problem (Schönemann, 1966) by singular value decomposition (SVD) and further refine the projection matrix using adaptive reranking loss. Albeit superior adaptation performance is obtained, we advocate that the complex-phased optimization can be substituted by end-to-end supervised learning with a lightweight adapter, which effortlessly provides comparable results given labeled image features.

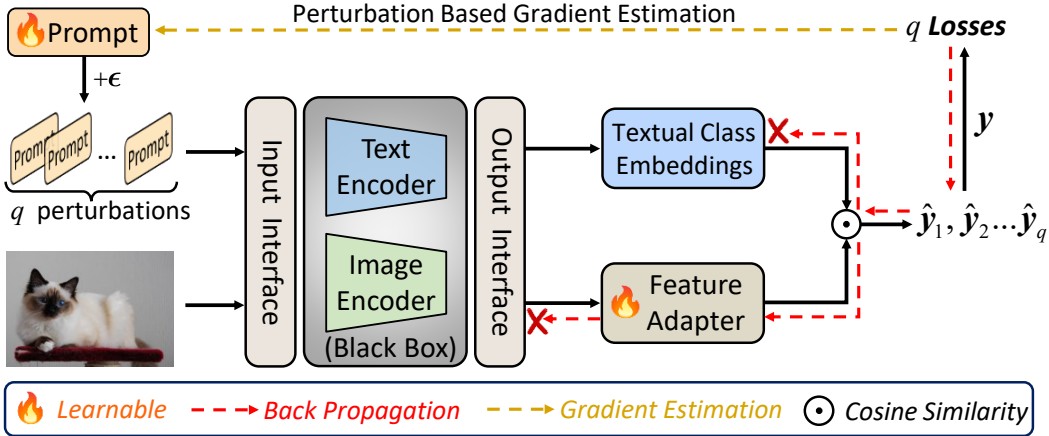

Figure 1: Overview of our proposed method. We collaboratively optimize the textual prompt and the image feature adapter for the adaptation of black-box pre-trained VL models. The prompt is optimized by estimated gradients since backpropagation cannot be applied to the black-box model. The visual adapter module is learned by direct supervised learning given output features from the pre-trained model.

## 3 Method

### 3.1 PEFT in the Black-box Framework

Here we introduce the general form of prompt tuning and adapter method and the dilemma when applied to black-box VL models.

**Prompt tuning for VL models.** Given a pre-trained VL model, *e.g.*, CLIP (Radford et al., 2021), existing soft prompt tuning approaches (Zhou et al., 2022b,a; Sun et al., 2022c) for classification tasks typically prepend learnable embeddings to the class names of the target dataset:

$$\phi(c_i) = [\boldsymbol{v}^1, \ldots, \boldsymbol{v}^M, \boldsymbol{c}_i] \qquad (1)$$

where $i \in \{1, \ldots, C\}$ denotes the index of classes, $\boldsymbol{c}_i$ denotes word embedding of the $i$-th class name $c_i$. For $j \in \{1, \ldots, M\}$, $\boldsymbol{v}^j$ is a learnable word embedding whose dimension is the same as the dimension of normal word embeddings in the vocabulary. The prediction of an input image $\boldsymbol{x}$ is obtained by computing similarities between the image feature $\boldsymbol{f}$ and prompted textual class features $\{\boldsymbol{t}_i\}_{i=1}^C$:

$$P(\hat{y} = i | \boldsymbol{x}; \boldsymbol{\phi}) = \frac{\exp(\langle \boldsymbol{t}_i, \boldsymbol{f} \rangle / \tau)}{\sum_{j=1}^C \exp(\langle \boldsymbol{t}_j, \boldsymbol{f} \rangle / \tau)} \qquad (2)$$

where the features of images are encoded by pre-trained image encoder $\boldsymbol{f} = \mathrm{Enc_I}(\boldsymbol{x})$, and textual class embeddings are generated by text encoder $\boldsymbol{t}_i = \mathrm{Enc_T}(\boldsymbol{\phi}(c_i))$. $\langle \cdot, \cdot \rangle$ calculates the cosine similarity and $\tau$ is a temperature parameter.

The objective of prompt module $\phi$ is maximizing the classification probability of the ground-truth

class of few-shot image samples:

$$\begin{aligned} \phi^* &= \arg\min_{\phi} \mathcal{L}(y, \boldsymbol{x}, \boldsymbol{\phi}) \\ &= \arg\min_{\phi} -\log P(\hat{y} = y | \boldsymbol{x}; \boldsymbol{\phi}) \end{aligned} \qquad (3)$$

When given a while-box model, it is straightforward to calculate the gradient of with respect to the prompt, and optimization of the prompt can be performed via gradient descent:

$$\phi_{t+1} = \phi_t - \eta_t \nabla_\phi \mathcal{L}(y, \boldsymbol{x}, \boldsymbol{\phi}) \qquad (4)$$

Unfortunately, in the black-box setting, the gradients are unable to be backpropagated through the pre-trained black-box $\mathrm{Enc_I}$ and $\mathrm{Enc_T}$ via the chain rule, and the term $\nabla_\phi \mathcal{L}(y, \boldsymbol{x}, \boldsymbol{\phi})$ cannot be directly obtained. Thus, current gradient-based prompt tuning methods are not feasible in this situation.

**Adapter learning for VL models.** Adapter learning methods (Gao et al., 2021; Zhang et al., 2022) for VL models usually manipulate the output features of pre-trained models for adaptation to target tasks. For instance, an adapter module can be introduced to transfer the visual features to new domains with $\hat{\boldsymbol{f}} = \boldsymbol{\psi}(\boldsymbol{f})$, and then the prediction is obtained by:

$$P(\hat{y} = i | \boldsymbol{x}; \boldsymbol{\phi}) = \frac{\exp(\langle \boldsymbol{t}_i, \hat{\boldsymbol{f}} \rangle / \tau)}{\sum_{j=1}^C \exp(\langle \boldsymbol{t}_j, \hat{\boldsymbol{f}} \rangle / \tau)} \qquad (5)$$

Learning such an adapter module by minimizing $\mathcal{L}(y, \boldsymbol{f}, \boldsymbol{\psi})$ does not require back-propagation through the entire pre-trained VL model, which provides convenience for adaptation without knowing the details of the backbone model. But access

to the output features of the pre-trained model is required to construct and optimize the adapter module (Zhang et al., 2022; Ouali et al., 2023).

**Further Analyses of the Black-box PEFT.** Given a black-box pre-trained model, the unavailability of gradients set a barrier to prompt tuning. Therefore, we intuitively have the idea of optimizing the prompt by estimating gradients. Input gradient approximation has been explored in the application of black-box model attacks (Ilyas et al., 2018b,a) and black-box model reprogramming (Tsai et al., 2020). We employ a perturbation-based gradient approximation method to estimate the gradient of learnable parameters in the prompt. The estimated gradient serves as an effective guide for the tuning of the prompt.

Although the gradient approximation technique provides barely satisfactory optimizing guidance, it is still suboptimal compared to the real gradients. Merely conducting single-step gradient descent based on the results of the estimated gradient leads to inefficient training. Inspired by the previous design of optimizers, we try to expedite the optimization based on the estimated gradient with a momentum. The basic idea is that information from previous updates is useful for the current step, and accumulated gradients possibly provide more promising exploration directions. we empirically find that equipping the momentum strategy for gradient approximation brings expedited convergence and remarkable adaptation performance gain.

Although we have no access to the internal variables of typical black-box models, under the circumstance where output features of the pre-trained VL backbone are available, post-processing adapter modules can be directly learned by labeled samples for PEFT.

Motivated by the above analyses, we propose to adapt black-box VL models with a collaborative PEFT consisting of optimization from two perspectives. Firstly, we tune a textual prompt under the guidance of the estimated gradient. Perturbation-based gradient approximation and effective optimization strategy are used to facilitate the training. Secondly, we learn a lightweight adapter to transfer pre-trained visual features. Joint optimization of the prompt and adapter brings superior adaptation performance. The overview of the proposed model is illustrated in Fig. 1.

In the following, we begin by presenting the perturbation-based gradient approximation method

in Section 3.2. Then, we explain how to expedite the tuning process by leveraging information from previous updates to achieve a better optimization in Section 3.3. Finally, we introduce the adapter module and joint training schedule in Section 3.3.

## 3.2 Perturbation Based Gradient Approximation

Suppose the prompt module $\phi$ has parameter $\boldsymbol{\theta}$ with dimension $D$. Let $f(\boldsymbol{\theta})$ be the loss function defined in Eq. (3). To approximate the gradient of the loss function with respect to $\boldsymbol{\theta}$, one possible avenue is to add a small increment to each dimension of $\boldsymbol{\theta}$ and sum up the slope of all dimensions:

$$\boldsymbol{g} = \sum_{i=1}^{D} \frac{f(\boldsymbol{\theta} + \beta \boldsymbol{e}_i) - f(\boldsymbol{\theta})}{\beta} \boldsymbol{e}_i \qquad (6)$$

where $e_i$ is a one-hot vector and its $i$-th element is equal to 1. Such an approximation may work well for low-dimensional parameters but is not suitable for problems where $D$ might be large. For example, the dimension of each word embedding of pre-trained CLIP is 512, *i.e.*, $\boldsymbol{\theta} \in \mathbb{R}^{M \times 512}$. Thus $M \times 512$ independent API calls for the black-box model must be applied to obtain the complete estimated gradient of parameter $\boldsymbol{\theta}$, which causes inefficiency.

To alleviate the cost of the above gradient estimation method, we adopt a stochastic perturbation-based gradient estimation technique formulated as:

$$\boldsymbol{g}_i = b \cdot \frac{f(\boldsymbol{\theta} + \beta \boldsymbol{\epsilon}_i) - f(\boldsymbol{\theta})}{\beta} \cdot \boldsymbol{\epsilon}_i \qquad (7)$$

$\boldsymbol{g}_i$ is the slope of the loss function along the direction of the perturbation. $\boldsymbol{\epsilon}_i$ is a vector randomly drawn from a unit sphere with an L2-norm of 1. $\beta$ is a small value controlling the scale of perturbations. $b$ is a scaling factor balancing the bias and variance trade-off of the estimator.

To mitigate noise in the estimated gradients, we sample random perturbation $\boldsymbol{\epsilon}_i$ for $q$ times, and the gradient of $\boldsymbol{\theta}$ is approximated by averaging the slope of $q$ directions (Wierstra et al., 2014; Ilyas et al., 2018a; Tu et al., 2019):

$$\overline{\boldsymbol{g}} = \frac{1}{q} \sum_{i=1}^{q} \boldsymbol{g}_i \qquad (8)$$

The upper bound of the estimation $\overline{\boldsymbol{g}}$ w.r.t. the true gradient $\nabla f(\boldsymbol{\theta})$ is analyzed in Tu et al. (2019)'s

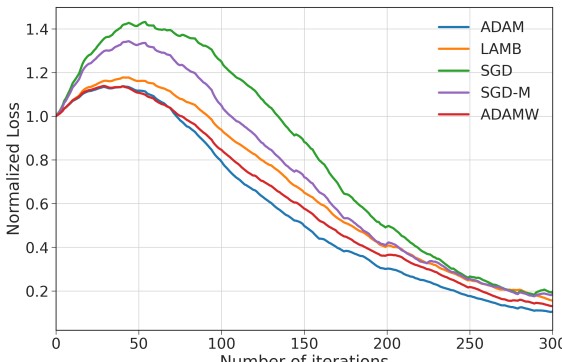

Figure 2: Trend of loss during training on EuroSAT. We adopt ADAM optimizer for expedited convergence and superior adaptation performance.

paper as:

$$\mathbb{E} \left\| \overline{\boldsymbol{g}} - \nabla f(\boldsymbol{\theta}) \right\|_2^2 \leq 4 \left( \frac{b^2}{D^2} + \frac{b^2}{Dq} + \frac{(b-D)^2}{D^2} \right) \cdot \left\| \nabla f(\boldsymbol{\theta}) \right\|_2^2 + \frac{2q+1}{q} b^2 \beta^2 L^2 \quad (9)$$

Setting a smaller $\beta$ can reduce the last error term in Eq. (9) but may cause an increase in noise due to numerical precision. Increasing the number of samples $q$ reduces the first error term but consumes more queries for the model API.

### 3.3 Effective Optimization Based on Estimated Gradient

To expedite the optimization based on the estimated gradient, we facilitate the tuning process by leveraging the momentum strategy. Specifically, we estimate the first-order moments of the parameters' gradient by $\boldsymbol{m}_t = \beta_1 \cdot \boldsymbol{m}_{t-1} + (1 - \beta_1) \cdot \overline{\boldsymbol{g}}_t$. The first-order moments accelerate the optimization and reduce the noise in the gradient of each step. And we obtain the adaptive estimation of the second-order moment by $\boldsymbol{v}_t = \beta_2 \cdot \boldsymbol{v}_{t-1} + (1 - \beta_2) \cdot \overline{\boldsymbol{g}}_t^2$, which is used to adjust the learning rate of each dimension adaptively.

In our experiments, we use optimizers that integrate the momentum as a practical implementation. To analyze the optimization results of different optimizers, we illustrate the trend of normalized loss value $|\mathcal{L}(\boldsymbol{\theta}^*) - \mathcal{L}(\boldsymbol{\theta})| / |\mathcal{L}(\boldsymbol{\theta}^*) - \mathcal{L}(\boldsymbol{\theta}_0)|$ in Fig. 2. Adam (Kingma and Ba, 2014) shows a fast and steady convergence and satisfied final results. We have also tried more advanced techniques, *e.g.*, LAMB (You et al., 2019), but no significant improvement in performance is observed. Empirical results show that optimizing the prompt with Adam optimizer based on the estimated gradient provides

expedited convergence and superior adaptation performance.

### 3.4 Visual Adapter Module

The pre-trained VL models can be effectively adapted to downstream tasks through the black-box prompt tuning method mentioned above. Meanwhile, under the assumption that having access to the output features of the black-box model (Ouali et al., 2023), a lightweight adapter module can be directly learned from labeled few-shot samples.

Adapter modules (Houlsby et al., 2019; Gao et al., 2021; Zhang et al., 2022) have been proven to be effective in the adaptation of VL models. During the training process of the adapter, the gradients do not need to be back-propagated through the entire pre-trained model, making it possible to equip the adapter module with black box models of which only the output features are available.

The text features have been adapted in our method by tuning the learnable prompt. Thus, we introduce an adapter module only for the visual features to achieve a collaborative adaptation. Specifically, we add an adapter module to the output of the visual encoder of the pre-trained VL model. Access to computed image features and labels allows the adapter to be learned at ease through direct supervised learning. During training, the visual adapter module and text prompts are optimized in turn to achieve a joint adaptation.

In our experiment, we attempt two simple but effective adapter designs, CLIP-Adapter (Gao et al., 2021) and Tip-Adapter (Zhang et al., 2022). Both of which can be well suited for the manipulation of image features for better adaptation.

## 4 Experiments

### 4.1 Implementation Details

**Datasets.** We perform the few-shot adaptation on black-box pre-trained CLIP (Radford et al., 2021) for image classification tasks following the general protocol in existing methods (Zhou et al., 2022b; Ouali et al., 2023; Oh et al., 2023). In particular, we adopt 11 commonly used datasets to evaluate our method, including ImageNet (Deng et al., 2009), Caltech101 (Li et al., 2004), Oxford-Pets (Parkhi et al., 2012), StanfordCars (Krause et al., 2013), Flowers102 (Nilsback and Zisserman, 2008), Food101 (Bossard et al., 2014), FGV-CAircraft (Maji et al., 2013), SUN397 (Xiao et al., 2010), UCF101 (Soomro et al., 2012), DTD (Cim-

Table 1: Few-shot adaptation performance on 11 image classification tasks. Black-box methods are indicated with gray shadows.

| Model | Method | Pets | Flowers | FGVCA | DTD | EuroSAT | Cars | Food101 | SUN397 | Caltech | UCF | ImageNet | Avg. |
|---|---|---|---|---|---|---|---|---|---|---|---|---|---|
| | CoOp (1 ctx) | 89.9 | 85.4 | 25.7 | 61.5 | 80.2 | 60.8 | 79.6 | 66.9 | 91.9 | 72.0 | 62.9 | 70.6 |
| RN50 | CLIP-Adapter | 87.9 | 94.7 | 33.0 | 67.4 | 85.9 | 69.4 | 77.0 | 68.0 | 92.5 | 77.8 | 60.4 | 74.0 |
| | Tip-Adapter | 89.7 | 95.2 | 37.5 | 67.6 | 84.4 | 74.7 | 78.9 | 70.6 | 93.1 | 78.3 | 63.9 | 75.8 |
| | ZSCLIP | 85.8 | 66.1 | 17.3 | 42.3 | 37.6 | 55.6 | 77.3 | 58.5 | 86.3 | 61.5 | 58.2 | 58.8 |
| | LFA | 86.8 | 94.6 | 35.9 | 66.4 | 84.1 | 73.6 | 76.3 | 71.3 | 92.7 | 77.0 | 63.7 | 74.8 |
| | Ours (w/o Adapter) | 89.2 | 83.8 | 23.8 | 60.9 | 77.3 | 59.4 | 79.6 | 65.1 | 91.6 | 69.6 | 62.3 | 69.3 |
| | Ours (CLIP-Adapter) | 88.0 | 94.9 | 35.6 | 67.4 | 85.1 | 73.8 | 77.3 | 68.5 | 93.1 | 78.6 | 63.8 | 75.1 |
| | Ours (Tip-adapter) | 89.9 | 95.2 | 37.3 | 68.4 | 85.3 | 74.5 | 78.5 | 71.0 | 92.6 | 79.4 | 64.6 | 76.1 |
| | CoOp (1 ctx) | 93.5 | 91.6 | 33.1 | 66.1 | 85.3 | 71.4 | 87.3 | 72.0 | 95.7 | 79.8 | 71.0 | 77.0 |
| | CoCoOp (1 ctx) | 92.8 | 86.7 | 31.4 | 61.7 | 73.8 | 68.9 | 87.1 | 71.6 | 94.8 | 77.4 | 70.5 | 74.2 |
| ViT/B16 | CoCoOp (4 ctx) | 92.9 | 85.8 | 31.4 | 61.9 | 72.5 | 68.3 | 87.3 | 72.2 | 94.9 | 77.1 | 71.0 | 74.1 |
| | CLIP-Adapter | 92.1 | 97.2 | 43.3 | 72.7 | 89.0 | 79.0 | 85.8 | 74.3 | 96.3 | 84.2 | 70.0 | 80.4 |
| | Tip-Adapter | 93.3 | 97.5 | 46.8 | 73.7 | 88.3 | 83.9 | 87.5 | 76.1 | 95.8 | 84.3 | 71.8 | 81.7 |
| | ZSCLIP | 89.2 | 71.3 | 24.7 | 44.4 | 47.6 | 65.3 | 86.1 | 62.5 | 92.9 | 66.8 | 66.7 | 65.2 |
| | BlackVIP | 89.7 | 70.6 | 25.0 | 45.2 | 73.1 | 65.6 | 86.6 | 64.7 | 93.7 | 69.1 | 67.1 | 68.2 |
| | LFA | 92.4 | 96.8 | 46.0 | 71.9 | 87.3 | 82.2 | 87.1 | 76.7 | 96.2 | 84.0 | 72.6 | 81.2 |
| | Ours (w/o adapter) | 93.7 | 88.6 | 30.7 | 64.0 | 81.0 | 68.9 | 87.2 | 71.1 | 95.8 | 78.8 | 70.6 | 75.5 |
| | Ours (CLIP-Adapter) | 92.2 | 97.2 | 45.3 | 73.3 | 88.8 | 81.2 | 86.1 | 74.8 | 95.8 | 84.6 | 71.9 | 81.0 |
| | Ours (Tip-Adapter) | 93.8 | 97.8 | 46.6 | 74.1 | 88.3 | 83.5 | 87.3 | 75.9 | 95.9 | 84.9 | 72.4 | 81.9 |

poi et al., 2014), and EuroSAT (Helber et al., 2019). For each dataset, labeled few-shot samples from each class are used as training data.

**Learnable Prompts.** The learnable prompts are shared across all classes in the target dataset. By default, the length of the prompt is set to be $M = 1$, which reduces the number of parameters in the learnable prompt. A small parameter optimization space helps maintain the quality of the estimated gradients with limited resource for exploration, resulting in effective tuning results. The effect of different prompt sizes is analyzed in Sec. 4.4. To initialize the prompt with different length, we use "a", "a photo", "a photo of a", and "a photo of a a photo of a" for $M = 1, 2, 4, 8$, respectively.

**Adapter Module.** Following CLIP-Adapter (Gao et al., 2021), our adaptor module adopts a two-layer MLP that follows the pre-trained visual encoder. The input and output dimensions are the same as the dimension of the CLIP image feature, and the number of hidden units is a quarter. Following Tip-Adapter (Zhang et al., 2022), we use the averaged feature of random augmented training images from 10 epochs as the initialization of the cache to construct the projection layer.

**Training Details.** We employ the official CLIP model to evaluate our proposed method. For a comprehensive comparison, we conduct experiments with different visual backbones, *i.e.*, ResNet50 and ViT/B16. The query number $q$ is set as $q = 256$ by default, and its effect is discussed in Sec. 4.4. The hyperparameters $b$ and $\beta$ in Eq. (7) are set as $D$ and $1/D$, respectively. $D$ is the dimension of the parameter in the prompt.

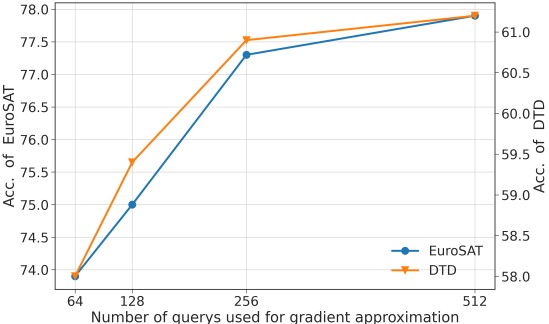

Figure 3: Ablation results of "Ours (w/o Adapter)" with different $q$.

## 4.2 Few-shot Adaptation Performance on Image Classification

We conduct extensive experiments on 11 datasets to evaluate our proposed method. Table 1 reports the 16-shot adaptation performance of competing methods. For a comprehensive comparison, we include both white-box PEFT methods (*i.e.*, CoOp (Zhou et al., 2022b), CoCoOp (Zhou et al., 2022a), CLIP-Adapter (Gao et al., 2021), and Tip-Adapter (Zhang et al., 2022)) and black-box methods (*i.e.*, Black-VIP (Oh et al., 2023) and LFA (Ouali et al., 2023)). "ZSCLIP" denotes the outcomes obtained using manually designed hard prompts.

From Table 1, our black-box prompt tuning method (ViT/B16 backbone) surpasses previous work Oh et al. (2023) with an average accuracy margin of 7.3% across 11 datasets, demonstrating the effectiveness of our black-box textual prompting for the adaptation of the VL model. Furthermore, when the context length of the prompt is fixed as $M = 1$, our black-box prompt tuning method performs comparably to the white-box prompt method,

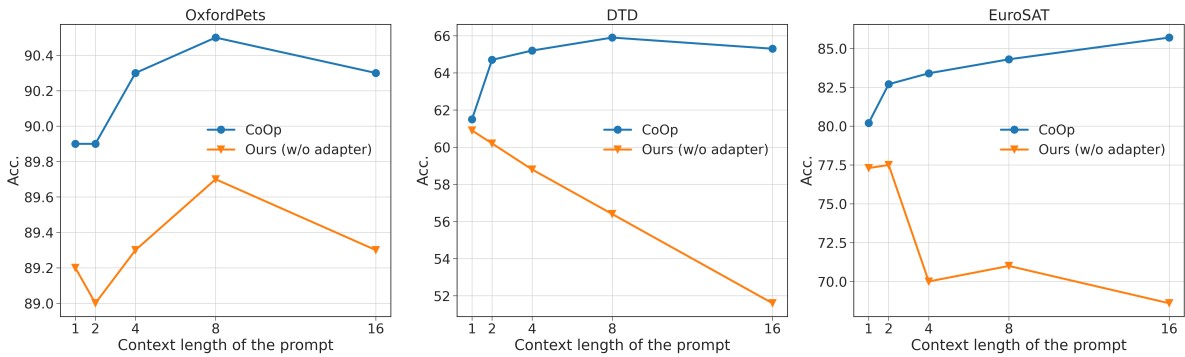

Figure 4: Ablation results of "Ours (w/o Adapter)" with different context length of the prompt.

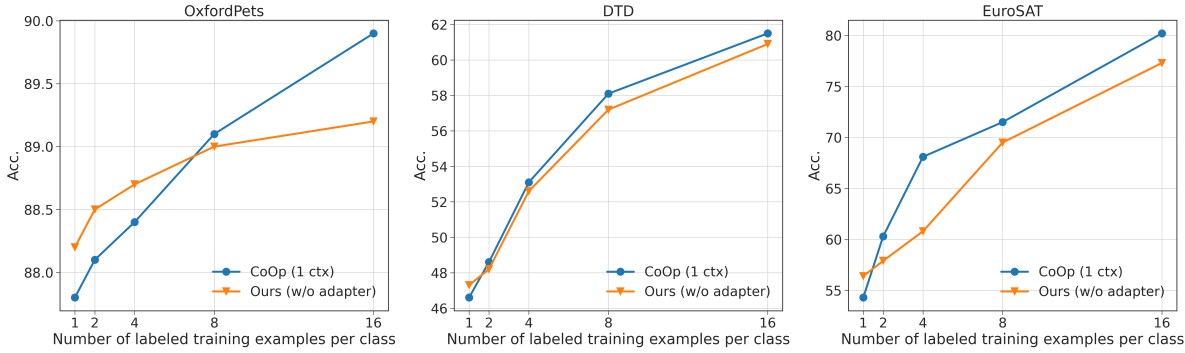

Figure 5: Ablation results of "Ours (w/o Adapter)" with different quantity of few-shot training data.

Table 2: Comparison of different black-box optimizers.

| Method | API calls / iter | EuroSAT Acc. |
|---|---|---|
| CMA-ES | 10 | 61.9 |
| SPSA-GC | 10 | 60.1 |
| Ours (w/o adapter) | 10 | **62.6** |

*i.e.*, CoOp (1 ctx), with a slight difference of less than 2%.

By assuming pre-computed features are available, LFA (Ouali et al., 2023) optimizes a projection layer in a multi-stage procedure as introduced in Section 2. We advocate that end-to-end learning of adapter methods (Gao et al., 2021; Zhang et al., 2022) provides a much more brief avenue meanwhile gives satisfactory performance. As shown in Table 1, optimizing the adapter module from CLIP-Adapter and Tip-Adapter can achieve comparable performance with LFA. Thus, we integrate our black-box prompt tuning method with these more flexible adapter modules. From Table 1, the collaborative adaptation of black-box prompting and adapter module brings remarkable performance and achieves a new state-of-the-art result.

### 4.3 Comparison with Black-Box Optimizers

Existing black-box prompt tuning methods have explored various effective optimization techniques when the gradient is unavailable. Here we compare our method with two other different optimization algorithms based on our implementation. In particular, CMA-ES algorithm (Hansen et al., 2003) is considered as state-of-the-art in evolutionary computation and is previously used to optimize the prompt for large language models (Sun et al., 2022b,a). SPSA-GC was proposed by BlackVIP (Oh et al., 2023) to learn a visual prompt for adaptation of pre-trained CLIP.

For a fair comparison, we unify the number of API calls per iteration for all competitors to 10. This is achieved by: setting the population size of CMA-ES as 10; setting the number of repeated two-side estimations of SPSA-GC as 5; setting the number of samplings of our perturbation-based gradient approximation as $q = 10$. The experiments are conducted on CLIP ResNet50 model, and the prompt length was set to 1. All optimizers are trained for 750 iterations until convergence, and the results are listed in Table 2. From the Table, our method outperforms the SPSA-GC algorithm, which is also based on gradient estimation. Although CMA-ES exhibits faster convergence, noticeable fluctuations are observed even in the later stages of training. Our perturbation-based gradient approximation method is more suitable for the adaption of the VL model.

## 4.4 Ablation Study

Ablation studies are performed to evaluate the effect of various factors, including the number of queries, the prompt length, the number of few-shot samples, and the collaborative training schedule. The experiments are mainly on the CLIP ResNet50 model.

**Effect of the number of queries** $q$**.** The number of samplings $q$ controls the times of querying the black-box model in each iteration. It has a significant impact on the number of API calls required for learning the prompt. Fig. 3 illustrates the adaptation performance with different $q$ values. Generally, larger values of $q$ yield more reliable gradients but also require more time and API calls for the black-box model. To trade-off the performance and computational cost, we use $q = 256$ for the results presented in Section 4.2.

**Effect of prompt length.** We further investigate the effect of Prompt length $M$. For comparison, all the experiments are conducted under 16-shot training data, with the same number of sampling ($q = 256$) and iterations. The results are illustrated in Fig. 4. One can see that the trend of performance on different tasks varies as the context length of the prompt changes. For white-box prompt tuning, longer prompts usually can lead to better adaptation to downstream datasets, *e.g.*, DTD and EuroSAT. However, blindly lengthening the context (*e.g.* $M = 16$) will not result in continuously rising performance. Increasing the length of context brings little improvement for OxfordPets. We attribute these results to the varying degrees of data diversity among different tasks.

But in the case of black-box models, the experimental phenomenon changes due to the influence of gradient approximation. Lengthening the context of the prompt brings trivial benefits and may even result in noticeable performance degradation. The expanded parameter space of a long context leads to practical difficulties in gradient estimation thus the optimization may lead to a suboptimal result. Increasing the number of sampling $q$ may improve the reliability of estimated gradients, but scaling up $q$ in proportion to the size of the prompt leads to severe inefficiency. Thus, we use the prompt length of 1 as a trade-off.

**Effect of the number of few-shot samples.** The number of few-shot samples determines the amount of training data used to adapt the pre-trained VL model. To demonstrate its effect, we keep the default configuration and vary the number of samples used for prompt tuning. Both black box and white box models undergo the same number of iterations. As shown in Fig. 5, increasing the number of samples clearly leads to better adaptation results. Moreover, we observe that in extremely data-scarce scenarios with only 1-shot sample per class, tuning the prompt based on the estimated gradient outperforms white-box tuning on all three datasets. One possible explanation is that optimizing with true gradients can lead to overfitting when the amount of data is too small. In contrast, gradient approximation provides a more robust optimization direction. As the amount of data increases, the advantages of direct white-box learning become more obvious.

Table 3: Ablation study on the training schedule.

| Datasets | P-A | A-P | ALT |
|---|---|---|---|
| EuroSAT | 84.0 | 82.7 | **85.1** |
| DTD | 66.4 | 65.7 | **67.4** |
| Caltech101 | 91.9 | 92.7 | **93.1** |
| OxfordPets | 87.9 | 87.5 | **88.0** |

**Effect of the collaborative training schedule.** In our experiment, the prompt and the adapter module are optimized jointly to maximize their collaborative performance. During training, we alternately update the prompt and the adapter module at different epochs. To assess the effectiveness of this joint optimization schedule, we conducted experiments using three different ways of training: (i) tuning the prompt until convergence and then optimizing the adapter module (P-A); (ii) tuning the adapter module until convergence and then optimizing the prompt (A-P); (iii) our collaborative training schedule (ALT). We train "Ours (CLIP-Adapter)" under the above three schedules, and the results are shown in Table 3. As shown in the table, recurrently updating the prompt and the adapter alternately (ALT) achieves superior collaborative adaptation performance, demonstrating its effectiveness.

## 5 Conclusion

In this paper, we present CBBT, a black-box adaptation approach for VL models. We effectively tune a soft prompt for the text encoder by gradient approximation and jointly learn a lightweight adapter module to transfer the visual features of the pre-trained backbone. Equipped with the textual prompt and the visual adapter, our method achieves a collaborative adaptation for both modalities. Experiments on various datasets show that our CBBT performs favorably against the state-of-the-art methods.

## Limitations

We optimize the prompt in the original high-dimensional prompt embedding space, which leads to unsatisfactory optimization results for the prompt with a long context, as shown in Section 4.4. The high-dimensional parameter in the prompt also makes the gradient approximation more difficult. We have tried to optimize the prompt in a smaller subspace following the approach in BBT (Sun et al., 2022b). But the adaptation performance decreased a lot even though we only released a small proportion of the original dimensions. The intrinsic dimensionality property (Aghajanyan et al., 2020; Qin et al., 2021) for vision-language pre-trained models needs further investigation.

Besides, we optimize a continuous prompt with the need for the token embedding layer of pre-trained models. Learning a discrete prompt for the adaptation of VL models is worthy of exploration, considering that the discrete text prompt provides an explicit explanation, and discrete text inputs are more suitable for the invocation of the latest pre-trained model APIs with natural language inputs and/or outputs.

## Acknowledgements

This work was supported in part by National Key R&D Program of China under Grant No. 2020AAA0104500, and by the National Natural Science Foundation of China (NSFC) under Grant No. U19A2073.

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

## A  Generalization Ability of Black-Box Prompt

To evaluate the generalization ability of our method, we conducted experiments on the extensively evaluated domain shift benchmarks and base-to-new setting (training on samples from base classes, testing on samples from new classes) commonly used in studies for adaptation of CLIP.

**Generalization to other domains.** Following CoOp (Zhou et al., 2022b) and CoCoOp (Zhou et al., 2022a), we evaluate the transferability of the prompt learned from ImageNet to the three specially designed datasets. The results are shown in Table 4. Given the high variance inherent in these trials, the results are averaged over three random re-runs to ensure reliable comparisons.

Our prompt learned by black-box optimization performs better than CoOp with a clear margin. Moreover, compared to CoCoOp, which relies on input-conditioned prompts generated by a meta-network, our vanilla prompt demonstrates superior performance on two of the three benchmarks.

**Generalization from base to new classes.** Following CoCoOp (Zhou et al., 2022a), we split the classes of the target dataset into two sets. In the base-to-new setting, the methods are trained using data from base classes and tested separately on base and new classes to evaluate the generalization ability to unseen classes in training. The results are shown in Table 5.

While CoOp improves pre-trained CLIP on base classes, it fails grievously on novel classes. CoCoOp optimizes for each instance to gain more generalization over an entire task. Our method achieves comparable results to CoCoOp by tuning a single prompt with the black-box optimizer. Optimizing the prompt by estimated gradient avoids the trend of overfitting to training samples, thus making up the superior of our method on generalization ability to white-box prompt tuning.

## B  More Results with Longer Prompt

In Fig. 4 of our paper, we optimize prompts with different lengths under a fixed training time budget by setting the same number of samplings $q$ as 256 for gradient approximation. Such a setting ensures training efficiency but may lead to suboptimal results for longer prompts, resulting in a performance drop of longer prompts. To demonstrate this, we have conducted experiments in which the value is scaled proportionately according to the size of the prompt, and the results are reported in Table 6.

From the table, with sufficient training time available, proportionately scaling the samplings for tuning of the longer prompts achieves stable convergence and clear improvements (especially on EuroSAT). Nonetheless, our optimized prompts consistently outperform hand-crafted hard prompts of any length.

## C  Computational Time Budget

The added computation burden of our method compared to white-box prompting methods lies within the multiple samplings required by the gradient approximation. We provide the training duration linked to the tuning methods presented in Table 1 on the EuroSAT dataset in Table 7. All training procedures are conducted on a single 3090 GPU. We record the minutes used for complete training and divide the time by the number of trained epochs to ascertain the time per epoch. While the sampling process inevitably elongates the training period, the overall consumed time is acceptable.

## D  Analysis of the Error in Gradient Estimation

The upper bound of the error of gradient approximation is $4\|\nabla f(\boldsymbol{\theta})\|_2^2$ according to Eq. (9). It is a theoretical value obtained through multiple bounding steps in the proof. The actual estimation error of the gradient during training is much lower than the theoretical upper bound since the experiments are conducted on reasonably annotated datasets with pre-trained CLIP and properly initialized prompts. As the training proceeds, the value of the true gradient becomes small, making the error of the estimated gradient, bounded by the true gradient, become small simultaneously. Thus, the results of "Ours (w/o adapter)" are closely comparable to "CoOp (1 ctx)" in Table 1.

## E  Applying to Larger Black-Box Models

It is promising to apply our method to larger black-box models. In fact, there exist closed-sourced model APIs, e.g., GPT-3, that provide the feature extraction function. It is possible to adapt pre-trained models of this kind by transferring the extracted features. Additionally, inspired by recent discrete prompt tuning approaches in Maus et al. (2023); Wen et al. (2023), it is practically

Table 4: Comparison of manual and learned prompt in domain generalization. The prompts are learned on 16-shot data from ImageNet.

| Methods | Source | Target | | |
|---|---|---|---|---|
| | ImageNet | ImageNet-Sketch | ImageNet-A | ImageNet-R |
| Zero-Shot CLIP | 66.7 | 46.2 | 47.8 | 74.0 |
| CoOp (4 ctx) | 71.5 | 48.0 | 49.7 | 75.2 |
| CoCoOp (4 ctx) | 71.0 | **48.8** | 50.6 | 76.2 |
| Ours (w/o adapter) | 70.7 | 48.7 | **50.7** | **76.6** |

Table 5: Comparison of manual and learned prompt in the base-to-new generalization setting. The prompts are learned from 16 images per base class.

| Methods | OxfordPets | | | EuroSAT | | | DTD | | |
|---|---|---|---|---|---|---|---|---|---|
| | Base | New | H | Base | New | H | Base | New | H |
| Zero-Shot CLIP | 91.2 | 97.3 | 94.1 | 56.5 | 64.1 | 60 | 53.2 | 59.9 | 56.4 |
| CoOp (4 ctx) | 93.7 | 95.3 | 94.5 | 92.2 | 54.7 | 68.9 | 79.4 | 41.2 | 54.2 |
| CoCoOp (1 ctx) | 94.6 | 95.6 | 95.1 | 84.2 | 55.3 | 66.8 | 75.1 | 53.6 | 62.6 |
| CoCoOp (4 ctx) | 95.2 | 97.0 | 96.1 | 86.0 | 59.9 | 70.6 | 73.2 | 55.4 | 63.1 |
| Ours (w/o adapter) | 95.8 | 95.8 | 95.8 | 90.8 | 71.1 | 79.8 | 77.9 | 51.1 | 61.7 |

Table 6: More results with longer prompt and varying samplings $q$. "ctx" denotes the length of the prompt.

| Methods | ctx | $q$ | OxfordPets | DTD | EuroSAT |
|---|---|---|---|---|---|
| Zero-Shot CLIP | 1 | - | 80.7 | 38.2 | 31.1 |
| Ours (w/o Adapter) | 1 | 256 | 89.2 | 60.9 | 77.3 |
| Ours (CLIP-Adapter) | 1 | 256 | 88.0 | 67.4 | 85.1 |
| Ours (TIP-Adapter) | 1 | 256 | 89.9 | 68.4 | 85.3 |
| Ours (w/o Adapter) | 2 | 256 | 89.0 | 60.2 | 77.5 |
| Ours (w/o Adapter) | 2 | 512 | 90.0 | 62.4 | 77.7 |
| Ours (CLIP-Adapter) | 2 | 512 | 87.5 | 67.2 | 85.4 |
| Ours (TIP-Adapter) | 2 | 512 | 89.6 | 68.8 | 85.0 |
| Zero-Shot CLIP | 4 | - | 83.6 | 40.0 | 24.2 |
| Ours (w/o Adapter) | 4 | 256 | 89.4 | 58.8 | 70.0 |
| Ours (w/o Adapter) | 4 | 1024 | 89.5 | 62.2 | 79.6 |
| Ours (CLIP-Adapter) | 4 | 1024 | 88.6 | 66.9 | 85.3 |
| Ours (TIP-Adapter) | 4 | 1024 | 89.8 | 69.3 | 85.2 |
| Zero-Shot CLIP | 8 | - | 84.2 | 39.3 | 31.0 |
| Ours (w/o Adapter) | 8 | 256 | 89.4 | 55.6 | 75.3 |
| Ours (w/o Adapter) | 8 | 2048 | 89.7 | 61.6 | 81.7 |
| Ours (CLIP-Adapter) | 8 | 2048 | 88.4 | 66.8 | 85.5 |
| Ours (TIP-Adapter) | 8 | 2048 | 90.0 | 68.4 | 85.4 |

feasible to discretize the learned prompts by projecting the continuous embedding to discrete token space to support a broader range of black-box models that only allows discrete input, e.g., ChatGPT, Bard. Our research will persist in exploring more practical adaptation techniques for vision-language models.

Table 7: Comparison of training time budget.

| Methods | min / epoch | min / train |
|---|---|---|
| CoOp | 0.017 | 3.3 |
| CoCoOp | 0.120 | 1.2 |
| Ours (w/o Adapter) | 0.095 | 14.2 |
| Ours (CLIP-Adapter) | 0.051 | 7.7 |
| Ours (Tip-Adapter) | 0.054 | 8.1 |