# OpenReview forum: "Black-Box Tuning of Vision-Language Models with Effective Gradient Approximation"
_EMNLP/2023/Conference — EMNLP 2023 Findings_

### Official Review · Reviewer_ztat · 2023-07-26

**Soundness:** 3

**Excitement:**

3: Ambivalent: It has merits (e.g., it reports state-of-the-art results, the idea is nice), but there are key weaknesses (e.g., it describes incremental work), and it can significantly benefit from another round of revision. However, I won't object to accepting it if my co-reviewers champion it.

**Paper Topic And Main Contributions:**

This paper improves the gradient estimation method in black-box PEFT of a vision-language model.

**Questions For The Authors:**

- A. A few questions about the upper bound Eq.(8)
    - A1. In Eq.(9), if we set $b = D$, the upper bound is at least $4\nabla||f(\theta)||_2^2$. I wonder how such a loose upper bound guarantees the convergence of the optimization process.
    - A2. Could the authors provide the difference between the estimated and the real gradient during training?

- B. Why more learnable prompts lead to worse performance? This is not consistent with white-box methods such as CoOp, CoCoOp[1]. Does the new gradient estimation method have an impact on this? For example, it has a loose upper bound and cannot guarantee the convergence of the optimization.
    - B1. I am not convinced by the "expanded parameter space" explanation as the adapter has more parameters than the prompt.

- C. In Table 1, does LFA adopt adapters? If it does not use an adapter, it seems much better than the proposed method.

- D. Finally, I think the setting of such black-box tuning is weird. We can inject learnable prompts (continuous vector rather than text) into the model. We can also get the feature of the model. This is what we cannot do for ChatGPT, Bard, and GPT-4 (lines 057-059 in the paper). What is the meaning of such a task?


[1] https://github.com/KaiyangZhou/CoOp

**Reasons To Accept:**

- This paper improves the gradient estimation method in black-box PEFT of a vision-language model.

- They combine prompt tuning and adapter tuning in black-box PEFT and achieve pretty good performance.

**Reasons To Reject:**

- Lack of in-depth theoretical analysis about why adopting the new gradient estimation method.
- Lack of thorough experimental analysis of the method.
- No explanations about why this method achieves good performance with Tip-Adapter + Prompt.

**Reproducibility:**

4: Could mostly reproduce the results, but there may be some variation because of sample variance or minor variations in their interpretation of the protocol or method.

**Reviewer Confidence:**

3: Pretty sure, but there's a chance I missed something. Although I have a good feel for this area in general, I did not carefully check the paper's details, e.g., the math, experimental design, or novelty.

---

> ### Author Rebuttal · Authors · 2023-08-28
>
> We thank **Reviewer ztat** for the valuable suggestions. Below we respond to the questions point by point.
>
> **Q-A1. The convergence of the optimization process**
>
> Although the upper bound of the gradient approximation error is $4 \Vert \nabla f(\boldsymbol \theta) \Vert_2^2$ according to Eq. (9), the actual estimation error of the gradient during training is much lower than the theoretical upper bound, which makes "Ours (w/o adapter)" closely comparable in performance to "CoOp (1 ctx)" in Table 1.
>
> Kindly refer to **Q-A2** for more details.
>
> **Q-A2. The difference between the estimated and the real gradient during training**
>
> Due to the inability to present figures in the rebuttal, we report the L-2 norm of the error relative to the norm of the real gradient in the table below by averaging every 150 iterations during training on the EuroSAT dataset. It shows that the error in practice is much lower than the theoretical upper bound in Eq. (9), i.e., 4, in our paper.
>
> |Training iteration|$150$|$300$|$450$|$600$|$750$|
> |:----:|:----:|:----:|:----:|:----:|:----:|
> |$\Vert \overline{\boldsymbol g} -  \nabla f(\boldsymbol \theta) \Vert_2^2 / \Vert \nabla f(\boldsymbol \theta) \Vert_2^2$|1.13|0.81|0.77|0.77|0.76|
>
>
> **Q-B. Performance drop with respect to the prompt length**
>
> In Fig. 4 of our paper, we optimize prompts with different lengths under a fixed training time budget by setting the same number of samplings $q$ as 256 for gradient approximation. Such a setting ensures training efficiency but may lead to suboptimal results for longer prompts, resulting in a performance drop of longer prompts. Larger $q$ brings better estimation for gradients according to Eq. (9). To thoroughly investigate the performance of longer prompts, we have conducted experiments in which the value $q$ is scaled proportionately according to the size of the prompt, and the results are reported in the table below. From the table, with sufficient training time available, longer prompts achieve stable convergence and clear improvements (especially on EuroSAT), similar to the results in white-box CoOp and CoCoOp.
>
> |Methods|OxfordPets|DTD|EuroSAT|
> |:----:|:----:|:----:|:----:|
> |Ours w/o Adapter (1 ctx, $q=256$)|89.2|60.9|77.3|
> |Ours w/o Adapter (2 ctx, $q=256$)|89.0|60.2|77.5|
> |Ours w/o Adapter (2 ctx, $q=512$)|90.0|62.4|77.7|
> |Ours w/o Adapter (4 ctx, $q=256$)|89.4|58.8|70.0|
> |Ours w/o Adapter (4 ctx, $q=1024$)|89.5|62.2|79.6|
> |Ours w/o Adapter (8 ctx, $q=256$)|89.4|55.6|75.3|
> |Ours w/o Adapter (8 ctx, $q=2048$)|89.7|61.6|81.7|
>
> **Q-C. The details of LFA**
>
> Yes, LFA also adopts a linear layer as an adapter, which is analogous to ours. From Table 1 in the main paper, under fair comparison, our method performs favourably against LFA.
>
> **Q-D. The setting of black-box tuning**
>
> Our black-box prompt tuning method follows settings in BBT[r1], BBTv2[r2], BSL[r3], where black-box optimization has been observed to yield better prompts than white-box tuning. Meanwhile, we link prompting with the available output features of the model by introducing adapters for a comprehensive comparison with the existing black-box adaptation method LFA[r4]. Actually, there exist closed-sourced model APIs, e.g., GPT-3, that provide the feature extraction function. Inspired by recent discrete prompt tuning approaches in [r5, r6], it's practically feasible to discretize the learned prompts by projecting the continuous embedding to discrete token space to support a broader range of black-box models, e.g., ChatGPT, Bard. Our research will persist in exploring more practical adaptation techniques for vision-language models.
>
> [r1] Black-Box Tuning for Language-Model-as-a-Service
> [r2] BBTv2: Towards a Gradient-Free Future with Large Language Models
> [r3] Black-box Prompt Tuning with Subspace Learning
> [r4] Black Box Few-Shot Adaptation for Vision-Language models
> [r5] Black Box Adversarial Prompting for Foundation Models
> [r6] Hard Prompts Made Easy: Gradient-Based Discrete Optimization for Prompt Tuning and Discovery
>
> **Response to Other Concerns**
>
> **Q1. Why adopt the new gradient estimation method**
>
> Prompt tuning has been observed to enhance the adaptation to downstream tasks for white-box vision-language models. Existing prompt tuning methods typically employ gradient backpropagation for optimization. However, in the circumstances of black-box model, the unavailability of gradients set a barrier to prompt tuning. Therefore, we intuitively devise the idea of estimating the gradients for learnable prompts. We employ a perturbation-based gradient approximation algorithm to effectively steer the optimization of prompts, resulting in remarkable performance achievements.
>
> **Q2. Insufficient experimental analysis of the method.**
>
> We have provided more experimental results to support our method.
>
> * Firstly, we have presented the value of the error term in gradient estimation during the training process (kindly refer to **Q-A**). The result shows that the error in practice is much lower than the theoretical upper bound in Eq. (9), which ensures competitive performance of our method compared to white-box prompting methods.
> * Secondly, we have provided the results of an investigation on different prompt lengths  (kindly refer to **Q-B**), which shows stable performance gain when elongating the prompt with properly set samplings $q$.
> * Thirdly, we have deployed our method to extensively evaluated domain shift benchmarks and base-to-new settings (training on samples from base classes, testing on samples from new classes) in line with other studies (kindly refer to **Q1 of Review ya24**). The results demonstrate preferable generalizability of our method.
> * Furthermore, we have provided the training time budget of our method (kindly refer to **Q2 of Review ya24**).
> These results offer a more in-depth analysis of our method.
>
> **Q3. Why this method achieves good performance with Tip-Adapter + Prompt**
>
> Prompt and adapter are two non-conflicting ways of adaptation, i.e., one for textual prompt optimization and the other for output feature adaptation. As illustrated in papers [r1, r2], the adapter modules facilitate a straightforward adaptation of pre-trained models to the target dataset. For a fair comparison with LFA [r2], we have also employed an adapter in our method, and the collaborative optimization of prompt and adapter contribute to the overall improvements, surpassing the individual results of each. Besides, we have also explored the effects of different adaptors (i.e., CLIP-adapter and Tip-Adapter). From Table 1, Tip-Adapter utilizes the training few-shot samples in an effective cached design, and contributes more to our overall performance.
>
> [r1] CLIP-Adapter: Better Vision-Language Models with Feature Adapters
> [r2] Tip-Adapter: Training-free Adaption of CLIP for Few-shot Classification

---

### Official Review · Reviewer_ya24 · 2023-08-04

**Soundness:** 3

**Excitement:**

4: Strong: This paper deepens the understanding of some phenomenon or lowers the barriers to an existing research direction.

**Paper Topic And Main Contributions:**

This paper proposes Collaborative Black-Box Tuning (CBBT) for the optimization of textual prompts and the adaptation of output features in the context of a vision language model. The model here is considered as a black-box, meaning that users only have access to itsAPI  but not to the inner workings of the model (i.e., they cannot directly adjust the gradient flow or model parameters).

Their method is evaluated based on its performance and its convergence - that is, how quickly and effectively it finds optimal solutions.

The paper's results demonstrate that this black-box tuning approach can yield results that are comparable to, or even surpass, those achieved in a white-box setting (where full access to the model's internals is allowed) under certain conditions. This indicates the potential effectiveness of their proposed methodology for real-world applications where full access to the model's parameters and gradients might not be possible.

**Questions For The Authors:**

please refer to the reasons to reject.

**Reasons To Accept:**

1. This research paper introduces an efficient methodology for black-box tuning of pre-trained vision language models.
2. The proposed method demonstrated strong performance across 11 different datasets specifically used for fine-grained image classification.
3. The authors provide a viable solution to issues in previous research related to the black-box prompt tuning of pre-trained vision language models. They analyze their results from various perspectives, providing valuable insights into the black-box tuning process by comparing the white-box setting.

**Reasons To Reject:**

1. To provide the generalizability of the prompt optimization process, it might be beneficial to include situations with distribution shifts(e.g., https://arxiv.org/abs/1903.12261, https://arxiv.org/abs/2006.16241 …) This inclusion would be in line with other white-box setting research papers.

2.  Could you provide details about the computational budget required for your method, such as the training time?

3. The proposed method might yield better results with longer prompts if a more accurately structured prompt is used, rather than repeating "a photo of a". Have you considered this?

4. Could you identify the reasons why the classification performances are generally significantly lower without the use of an adapter?

**Reproducibility:**

4: Could mostly reproduce the results, but there may be some variation because of sample variance or minor variations in their interpretation of the protocol or method.

**Reviewer Confidence:**

4: Quite sure. I tried to check the important points carefully. It's unlikely, though conceivable, that I missed something that should affect my ratings.

---

> ### Author Rebuttal · Authors · 2023-08-28
>
> We thank **Reviewer ya24** for the valuable suggestions. Below we respond to the questions point by point.
>
> **Q1. Experiments on situations with distribution shifts**
>
> Thanks for the suggestion! Following CoCoOp, we have conducted experiments on extensively evaluated domain shift benchmarks, and the results are shown in the table below. Given the high variance inherent in these trials, the results are averaged over three random re-runs to ensure reliable comparisons. From the table, our method shows superior robustness to domain shifts.
>
> |Methods|Source|Target|Target|Target|
> |:----:|:----:|:----:|:----:|:----:|
> |    |ImageNet|ImageNet-Sketch|ImageNet-A|ImageNet-R|
> |Zero-Shot CLIP|66.7|46.2|47.8|74.0|
> |CoOp (4 ctx)|71.5|48.0|49.7|75.2|
> |CoCoOp (4 ctx)|71.0|**48.8**|50.6|76.2|
> |Ours (w/o adapter)|70.7|48.7|**50.7**|**76.6**|
>
> Besides, to evaluate the generalizability of our method, we have also deployed our methods to the prevalent base-to-new setting (training on samples from base classes, testing on samples from new classes) commonly used in studies for adaptation of CLIP. From the table below, our black-box prompt tuning yields comparable results to while-box CoOp on base classes while exhibiting significantly improved generalization performance to new classes (especially on EuroSAT and DTD datasets). Such an advantage of our approach can be attributed to less overfitting due to the utilization of the approximated gradient.  It is worth noting that CoCoOp sometimes achieves slightly superior performance than ours on new classes; however, it comes at a considerable cost of degradation of performance on the base classes.
>
> |Methods|OxfordPets|OxfordPets|OxfordPets|EuroSAT|EuroSAT|EuroSAT|DTD|DTD|DTD|
> |:----:|:----:|:----:|:----:|:----:|:----:|:----:|:----:|:----:|:----:|
> |    |Base|New|H|Base|New|H|Base|New|H|
> |Zero-Shot CLIP|91.2|97.3|94.1|56.5|64.1|60.0|53.2|59.9|56.4|
> |CoOp (4 ctx)|93.7|95.3|94.5|92.2|54.7|68.9|79.4|41.2|54.2|
> |CoCoOp (1 ctx)|94.6|95.6|95.1|84.2|55.3|66.8|75.1|53.6|62.6|
> |CoCoOp (4 ctx)|95.2|97.0|96.1|86.0|59.9|70.6|73.2|55.4|63.1|
> |Ours (w/o adapter)|95.8|95.8|95.8|90.8|71.1|79.8|77.9|51.1|61.7|
>
> **Q2. Details of the computational budget**
>
> The added computation burden of our method compared to white-box prompting methods lies within the multiple samplings required by the gradient approximation. We report the training durations linked to the tuning methods presented in Table 1 on the EuroSAT dataset in the table below. All the trainings are conducted on a single 3090 GPU. We record the minutes used for complete training and divide the time by the number of trained epochs to ascertain the time per epoch. While the sampling process inevitably elongates the training period, the overall consumed time is acceptable.
>
> |Methods|min / epoch|min / train|
> |:----:|:----:|:----:|
> |CoOp|0.017|3.3|
> |CoCoOp|0.120|1.2|
> |Ours w/o Adapter|0.095|14.2|
> |Ours CLIP-Adapter|0.051|7.7|
> |Ours Tip-Adapter|0.054|8.1|
>
> **Q3. The results of longer prompts**
>
> We have explored the implementation of more interpretable and semantically structured initializations for longer prompts, for instance, "a very realistic and detailed photo of a" for context length 8 (denoted as **"8 ctx structured"** in the table below). The prompts initialized with repeated "a photo of a" is denoted as **"8 ctx"**. From the table, more semantically structured initialization ("8 ctx structured, $q=256$" compared to "8 ctx, $q=256$") leads to improvements on the EuroSAT dataset but does not work on OxfordPets and DTD.
>
> Actually, the reason for the performance drop of longer prompts is the insufficient number of samplings $q$ for gradient approximation. To demonstrate this, we have conducted experiments in which the value $q$ is scaled proportionately according to the size of the prompt, and the results are reported in the table below. From the table, with sufficient training time available, longer prompts achieve stable convergence and clear improvements (especially on EuroSAT). Considering that the enhancement resulting from a longer prompt and larger $q$ is not significant, we limit the scale of prompt and $q$ to ensure elevated training efficiency.
>
> |Methods|OxfordPets|DTD|EuroSAT|
> |:----:|:----:|:----:|:----:|
> |Ours w/o adapter (1 ctx, $q=256$)|89.2|60.9|77.3|
> |Ours w/o adapter (4 ctx, $q=256$)|89.4|58.8|70.0|
> |Ours w/o adapter (4 ctx, $q=1024$)|89.5|62.2|79.6|
> |Ours w/o adapter (8 ctx, $q=256$)|89.7|56.4|71.0|
> |Ours w/o adapter (8 ctx structured, $q=256$)|89.4|55.6|75.3|
> |Ours w/o adapter (8 ctx structured, $q=2048$)|89.7|61.6|81.7|
>
>
> **Q4. The performance is lower without the adapter**
>
> Prompt and adapter are two non-conflicting ways of adaptation. As illustrated in papers [r1, r2], the adapter modules facilitate a straightforward adaptation of pre-trained models to the target dataset. For a fair comparison with LFA [r2], we have also employed an adapter to transfer output features. The collaborative optimization of prompt tuning and adapter learning contribute to the overall improvements, surpassing the individual results of each. From Table 1 of main paper, though our method shows performance drop without the adapter, our results still outperform existing state-of-the-art black box prompt tuning method (i.e., BlackVIP) by a large margin.
>
> [r1] Learning multiple visual domains with residual adapters
> [r2] Black Box Few-Shot Adaptation for Vision-Language models

---

### Official Review · Reviewer_QPix · 2023-08-04

**Soundness:** 3

**Excitement:**

3: Ambivalent: It has merits (e.g., it reports state-of-the-art results, the idea is nice), but there are key weaknesses (e.g., it describes incremental work), and it can significantly benefit from another round of revision. However, I won't object to accepting it if my co-reviewers champion it.

**Paper Topic And Main Contributions:**

This research paper introduces Collaborative Black-Box Tuning (CBBT), a new method for adapting large vision-language models to specific tasks. CBBT addresses the common issue of inaccessible model parameters by optimizing textual prompts and adapting output features for black-box models. The approach utilizes perturbed prompts to approximate gradients and incorporates a lightweight adapter for output feature adaptation. CBBT, tested across eleven benchmarks, shows significant improvement over existing black-box vision-language adaptation methods.

**Questions For The Authors:**

Please see the [reasons to reject] section.

**Reasons To Accept:**

The paper introduces a black-box prompt tuning method for pretrain VLM, such as CLIP, showing comparable performance with white-box prompt tuning across 11 different datasets. This could be beneficial when the parameters of the VLM is not available to the users.

**Reasons To Reject:**

1. I feel that paper has insufficiant baseline. For example, CoCoOp (https://arxiv.org/abs/2203.05557) is a widely used baseline for prompt tuning research in CLIP. Moreover, it would be nice to include the natural data shift setting as in most other prompt tuning papers for CLIP.

2. It would be nice to include the hard prompt baseline in Table 1 to see the increase in performance of each method.

3. I think the performance drop seen with respect to the prompt length (Figure 4) is a major limitation of this approach. For example, this phenomenon might make it so that using just a general hard prompt of length 4 ('a photo of a') would outperform the CBBT with length 4 or even CBBT with length 1.

**Reproducibility:**

5: Could easily reproduce the results.

**Reviewer Confidence:**

4: Quite sure. I tried to check the important points carefully. It's unlikely, though conceivable, that I missed something that should affect my ratings.

---

> ### Author Rebuttal · Authors · 2023-08-28
>
> We thank **Reviewer QPix** for the valuable suggestions. Below we respond to the questions point by point.
>
> **Q1. Comparisons with more baselines**
>
> We have conducted the comparisons with CoCoOp ([https://arxiv.org/abs/2203.05557](https://arxiv.org/abs/2203.05557)), and the results are reported in the below table. Specifically, we employ the official implementation of CoCoOp and train it with default training configuration, based on CLIP-ViT/B16, under the few-shot learning setting. As shown in the table, our method performs favourably against CoCoOp, demonstrating its effectiveness.
>
> |Method|Pets|Flowers|FGVCA|DTD|EuroSAT|Cars|Food101|SUN397|Caltech|UCF|ImageNet|Avg.|
> |:----|:----|:----|:----|:----|:----|:----|:----|:----|:----|:----|:----|:----|
> |CoOp (1 ctx)|93.5|91.6|33.1|66.1|85.3|71.4|87.3|72.0|95.7|79.8|71.0|77.0|
> |CoCoOp (1 ctx)|92.8|86.7|31.4|61.7|73.8|68.9|87.1|71.6|94.8|77.4|70.5|74.2|
> |CoCoOp (4 ctx)|92.9|85.8|31.4|61.9|72.5|68.3|87.3|72.2|94.9|77.1|71.0|74.1|
> |CLIP-Adapter|92.1|97.2|43.3|72.7|89.0|79.0|85.8|74.3|96.3|84.2|70.0|80.4|
> |Tip-Adapter|93.3|97.5|46.8|73.7|88.3|83.9|87.5|76.1|95.8|84.3|71.8|81.7|
> |ZSCLIP|89.2|71.3|24.7|44.4|47.6|65.3|86.1|62.5|92.9|66.8|66.7|65.2|
> |BlackVIP|89.7|70.6|25.0|45.2|73.1|65.6|86.6|64.7|93.7|69.1|67.1|68.2|
> |LFA|92.4|96.8|46.0|71.9|87.3|82.2|87.1|**76.7**|**96.2**|84.0|**72.6**|81.2|
> |Ours (w/o adapter)|93.7|88.6|30.7|64.0|81.0|68.9|87.2|71.1|95.8|78.8|70.6|75.5|
> |Ours (CLIP-Adapter)|92.2|97.2|45.3|73.3|**88.8**|81.2|86.1|74.8|95.8|84.6|71.9|81.0|
> |Ours (Tip-Adapter)|**93.8**|**97.8**|**46.6**|**74.1**|88.3|**83.5**|**87.3**|75.9|95.9|**84.9**|72.4|**81.9**|
>
> **Q2. Experiments under natural data shift setting**
>
> Thanks for the suggestion!  Following CoCoOp, we have conducted experiments on extensively evaluated domain shift benchmarks, and the results are shown in the table below. Given the high variance inherent in these trials, the results are averaged over three random re-runs to ensure reliable comparisons. From the table, our method also shows superior robustness to domain shifts.
>
> |Methods|Source|Target|Target|Target|
> |:----|:----|:----|:----|:----|
> |    |ImageNet|ImageNet-Sketch|ImageNet-A|ImageNet-R|
> |Zero-Shot CLIP|66.7|46.2|47.8|74.0|
> |CoOp (4 ctx)|71.5|48.0|49.7|75.2|
> |CoCoOp (4 ctx)|71.0|**48.8**|50.6|76.2|
> |Ours (w/o adapter)|70.7|48.7|**50.7**|**76.6**|
>
> Besides, to verify the generalizability of our method, we have also deployed our methods to the prevalent base-to-new setting (training on samples from base classes, testing on samples from new classes) commonly used in studies for adaptation of CLIP. From the table below, our black-box prompt tuning yields comparable results to while-box CoOp on base classes while exhibiting significantly improved generalization performance to new classes (especially on EuroSAT and DTD datasets). Such an advantage of our approach can be attributed to less overfitting due to the utilization of the approximated gradient.  It is worth noting that CoCoOp sometimes achieves slightly superior performance than ours on new classes; however, it comes at a considerable cost of degradation of performance on the base classes, which also elucidates the shortcomings of CoCoOp in the vanilla few-shot learning experiment.
>
> |Methods|OxfordPets|OxfordPets|OxfordPets|EuroSAT|EuroSAT|EuroSAT|DTD|DTD|DTD|
> |:----|:----|:----|:----|:----|:----|:----|:----|:----|:----|
> |    |Base|New|H|Base|New|H|Base|New|H|
> |Zero-Shot CLIP|91.2|97.3|94.1|56.5|64.1|60.0|53.2|59.9|56.4|
> |CoOp (4 ctx)|93.7|95.3|94.5|92.2|54.7|68.9|79.4|41.2|54.2|
> |CoCoOp (1 ctx)|94.6|95.6|95.1|84.2|55.3|66.8|75.1|53.6|62.6|
> |CoCoOp (4 ctx)|95.2|97.0|96.1|86.0|59.9|70.6|73.2|55.4|63.1|
> |Ours (w/o adapter)|95.8|95.8|95.8|90.8|71.1|79.8|77.9|51.1|61.7|
>
> **Q3. Comparison with the hard prompt baseline**
>
> We apologize for the unclear explanation in our paper. In fact, the ZSCLIP (which is the abbreviation of "Zero-Shot CLIP") method in Table 1 represents the method using the manually crafted hard prompt, e.g., "a photo of a". More carefully crafted prompt templates lead to marginal average score improvements on 11 datasets. Note that the learned prompts in our method are consistently better than hand-crafted hard prompts of any length.
>
> **Q4. Performance drop with respect to the prompt length**
>
> In Fig. 4 of our paper, we optimize prompts with different lengths under a fixed training time budget by setting the same number of samplings $q$ as 256 for gradient approximation. Such a setting ensures training efficiency but may lead to suboptimal results for longer prompts, resulting in a performance drop of longer prompts. To demonstrate this, we have conducted experiments in which the value $q$ is scaled proportionately according to the size of the prompt, and the results are reported in the table below. From the table, with sufficient training time available, longer prompts achieve stable convergence and clear improvements (especially on EuroSAT). Nonetheless, our optimized prompts consistently outperform hand-crafted hard prompts of any length.
>
> |Methods|OxfordPets|DTD|EuroSAT|
> |:----:|:----:|:----:|:----:|
> |Zero-Shot CLIP (1 ctx, "a")|80.7|38.2|31.1|
> |Ours w/o Adapter (1 ctx, $q=256$)|89.2|60.9|77.3|
> |Ours w/o Adapter (2 ctx, $q=256$)|89.0|60.2|77.5|
> |Ours w/o Adapter (2 ctx, $q=512$)|90.0|62.4|77.7|
> |ZSCLIP (4 ctx, "a photo of a")|83.6|40.0|24.2|
> |Ours w/o Adapter (4 ctx, $q=256$)|89.4|58.8|70.0|
> |Ours w/o Adapter (4 ctx, $q=1024$)|89.5|62.2|79.6|
> |ZSCLIP (8 ctx, "a very … photo of a")|84.2|39.3|31.0|
> |Ours w/o Adapter (8 ctx, $q=256$)|89.4|55.6|75.3|
> |Ours w/o Adapter (8 ctx, $q=2048$)|89.7|61.6|81.7|

---

### Meta-Review · Area_Chair_S4mp · 2023-09-19

**Recommendation:** 4

**Metareview:**

This paper introduces a black-box prompt tuning method for CLIP finetuning. The authors have added additional results and have done a nice job during the rebuttal. After rebuttal, two reviewers are happy about the paper, while one reviewer still shows concerns about accepting the paper.

One one hand, reviewers agree that the problem studied in this paper is interesting, and results are comprehensive. During rebuttal, additional results are added. There are a large attention recently towards black-box settings in language and vision-language domains and the proposed paper outperforms previous performance on black-box CLIP image classification setting, thus, reviewers recommend to accept the paper.

On the other hand, some reviewers think that the newly added experiments in the rebuttal derive different conclusions from the experiments in the main paper, i.e., whether a longer prompt works. Besides, a more thorough analysis of the upper bound is needed. Therefore, rejection is recommended. Also, some reviewer commented that the black-box tuning setting is kind of weird, where learnable prompts can be injected, and the feature of the model can be obtained.

---

### Decision · Program_Chairs · 2023-10-07

**Decision:**

Accept-Findings

**Comment:**

This paper introduces a black-box prompt tuning method for CLIP finetuning. The authors have added additional results and have done a nice job during the rebuttal. After rebuttal, two reviewers are happy about the paper, while one reviewer still shows concerns about accepting the paper.

One one hand, reviewers agree that the problem studied in this paper is interesting, and results are comprehensive. During rebuttal, additional results are added. There are a large attention recently towards black-box settings in language and vision-language domains and the proposed paper outperforms previous performance on black-box CLIP image classification setting, thus, reviewers recommend to accept the paper.

On the other hand, some reviewers think that the newly added experiments in the rebuttal derive different conclusions from the experiments in the main paper, i.e., whether a longer prompt works. Besides, a more thorough analysis of the upper bound is needed. Therefore, rejection is recommended. Also, some reviewer commented that the black-box tuning setting is kind of weird, where learnable prompts can be injected, and the feature of the model can be obtained.